# Image Segmentation of a Sewer Based on Deep Learning

Min He [1], Qinnan Zhao [1,*], Huanhuan Gao [2], Xinying Zhang [1] and Qin Zhao [1]

1   School of Civil Engineering and Architecture, Xi'an University of Technology, Xi'an 710048, China; hem@xaut.edu.cn (M.H.); zhangxinying@xaut.edu.cn (X.Z.); zhaoqin6688@xaut.edu.cn (Q.Z.)
2   PowerChina Northwest Engineering Corporation Limited, Xi'an 710048, China; huan8s@163.com
*   Correspondence: 2210720201@stu.xaut.edu.cn; Tel.: +86-139-5025-4142

**Abstract:** An accurate assessment of the type and extent of sewer damage is an important prerequisite for maintenance and repair. At present, distinguishing drainage pipe defect types in the engineering field mainly relies on the human eye, which is time consuming, labor intensive, and subjective. Some studies have used deep learning to classify the types of pipe defects, but this method can only identify one main pipe defect. However, sometimes a combination of defects, such as corrosion and precipitation on a section of pipe wall, can be classified as one category by picture classification, which is significantly different from the reality. Furthermore, the deep learning method for defect classification is unable to pinpoint the precise location and severity of a defect or estimate the number of flaws and the cost of maintenance and repair. Therefore, an image segmentation method based on deep convolutional neural networks is proposed to achieve pixel-level image segmentation of defect regions while classifying pipe defects. Compared with the deep learning network for defect classification, it can segment a variety of defects and reduce the number of samples, which is convenient for defect measurement. First, the image defect locations of seven typical defects were manually labeled to create the dataset. Then, a model based on the SegNet network was used to label defect areas automatically in an image. The pipeline image dataset was used to test the previously trained model using the CamVid dataset. Finally, the model was applied to drainage pipe network images that were provided by periscope and closed-circuit television inspection cameras, and the pixel accuracy of image segmentation reached 80%. From the results, it can be concluded that image segmentation and annotation technology based on deep learning is applicable to sewer defect detection. The identification results of pipeline defects were accurate. The SegNet model is a reliable method for image analysis of pipeline defects, which can accurately evaluate the type and degree of sewer damage.

**Keywords:** sewer defect; deep learning; image segmentation; SegNet network

## 1. Introduction

Sewer pipes are critical to the smooth operation of modern communities. On the one hand, they are an important infrastructural component and vital to a city's operations. On the other hand, their performance seriously affects the normal lives of citizens. Therefore, it is especially important to check existing drainage pipes regularly, to discover sewer defects in time, and to take reasonable and effective maintenance measures.

At present, closed-circuit television (CCTV) [1] is one of the most widely used pipeline detection methods. Although this method can obtain images of the inner wall of a sewer, it requires knowledgeable technicians to analyze the internal conditions, and this process is subjective and less automated. Image processing research in deep learning has advanced significantly in recent years, with significant progress made in a variety of domains. Some researchers have used deep learning to classify defects automatically, but most classifications based on deep learning are single-label classifications. A very recent study [2] provides a multilabel classification framework, but this classifier does not perform well enough. Moreover, determining specific information about the defect (such as location,

engineering quantity, etc.) remains difficult with artificial estimation methods and cannot be fully automated.

This paper proposes an automated defect segmentation and annotation method based on deep convolutional networks. The method uses a large amount of detection data to train a deep convolution network using the SegNet network as the backbone network. The resulting network was used to classify and segment various defects with image labels. The main objective was to meet the needs of large-scale sewage network survey work and to achieve intelligent defect detection. The results provide a basis and reference for further quantitative evaluation of sewer defects.

## 2. Related Work

Various methods exist for detecting the condition of sewers, with the most commonly used being CCTV. In recent years, sewer robots [3] have also been put into use in pipeline inspection. These methods can obtain high-quality sewer images. Therefore, many researchers have begun to study image-based automated defect detection methods. Two main methods are currently available: algorithm-based image processing technology and deep learning technology that relies on supervised learning.

### 2.1. Image Processing Methods

Zhun Fan et al. [4] overcame the limitations of established image processing-based crack detection methods and proposed a new automatic pavement crack detection and measurement method. Motamedi et al. [5] performed grayscale, filtering, and morphological processing on sewer images, which provided a certain theoretical basis for the nondestructive detection of urban drainage pipeline defects, although practical applications were more complicated. Kirstein et al. [6] detected drainage pipe defects using the shortest path algorithm, the Hough transform algorithm, and the Canny edge detector. This method combines optimal theory with traditional image processing technology to detect sewer defects. It has certain theoretical research significance, but the method is too cumbersome to be suitable for practical rapid-detection applications. Hawari et al. [7] proposed a complex algorithm combining image processing and shape analysis to distinguish sewer defects. As a result of the limitations of its dataset, the effectiveness of this method needs further determination.

### 2.2. Deep Learning Methods

Deep learning is an emerging method in the field of sewer inspection. In 1943, McCulloch and Pitts [8] first proposed the concept of artificial neural networks and mathematical models of artificial neurons. In 1989, LeCun [9] and colleagues presented the basic model of convolutional neural networks, which is one of the basic types of deep learning networks.

In 2006, Hinton and Salakhutdinov [10] first proposed the concept of deep learning and improved the neural network model algorithm. Ever since, deep learning technology has been developing rapidly, and deep learning algorithms have shown great advantages in the fields of voice, image, and natural language processing. In the 2012 ImageNet competition, Professor Hinton's students increased the depth of their convolutional neural network and used this model to achieve the highest score in the history of the competition. Ever since, deep learning has attracted more and more attention. After Long et al. [11] proposed the principle of fully convolutional networks (FCN), the classification objectives of deep convolutional neural networks extended from the object to the pixel and to the field of semantic segmentation. In 2015, Cambridge [12] proposed SegNet, an image semantic segmentation network designed to support autonomous driving or intelligent robots, which also achieved very good results in multiclass segmentation. Subsequently, semantic separation networks such as DeepLab [13] and PSPNet [14] came into being.

With the continuous development of deep learning, the deep convolutional neural network model began to be applied to detect sewer defects. Rahman [15] trained multiple deep neural networks to distinguish between single defects and normal images and achieved an

overall accuracy of 90%, but the dataset type was relatively simple. Sinha and Fieguth [16] proposed a neuro-fuzzy classifier that combined fuzzy logic testing with neural networks (ANN) to test the classification of defects in pipelines. Recently, Li et al. [17] proposed a hierarchical classification method for sewer defects based on deep convolutional neural networks, which is superior to traditional methods in classifying unbalanced datasets. At present, pipeline detection methods based on deep learning mainly include automatic classification methods. After the defect has been automatically classified, information, such as defect position and engineering quantity, still needs the judgment of professional technicians, meaning that the whole process of defect detection cannot be automated. Therefore, the image segmentation method based on deep learning is a new idea for sewer defect detection.

## 3. Image Segmentation

### 3.1. Comparison of Image Classification and Image Segmentation

Previously proposed automatic defect methods typically use feature extraction, resulting in poor generalization. Meijer D et al. [2] designed a convolutional neural network (CNN) to classify the sewer image set for defects and to automatically detect the 12 most common types of defects. Kumar et al. [18] developed an automatic defect classification framework using multiple binary CNNs to classify defects, such as root intrusion, deposits, and cracks, with an average detection accuracy of 86.2%. However, Kumar realized that he could not locate the defect, so he researched further. In 2020, Kumar [19] proposed another deep learning detection model to automatically classify and locate the defect location.

Image classification is the analysis of an image as a whole to classify it into one of several categories to replace human visual interpretation. Since an image may contain multiple objects belonging to different classes, the need to locate and identify them has led to further developments in image segmentation.

Image segmentation is a prediction of different features of an image: there may be multiple objects, people, or even multiple layers of background in an image, and people want to be able to predict which part of the image any pixel point belongs to. Image segmentation is the foundation of image processing. The accuracy of image segmentation directly influences the result of subsequent image processing. Therefore, obtaining good image segmentation results is more challenging and complex than image classification.

### 3.2. Image Segmentation Method

At present, there are thousands of image segmentation methods in the field of image processing. With the wide application of image segmentation methods and the rapid development of other related scientific fields, new image segmentation algorithm theories have emerged one after another. Different image segmentation algorithms each have their corresponding theoretical basis. The most common image segmentation algorithms are as follows: (1) threshold-based; (2) image segmentation based on edge detection; (3) region-based image segmentation; and (4) image segmentation based on neural network technology. Xu et al. [20] first used an image segmentation algorithm for the automatic detection of sewer defects. More recently, Sinha [21] used morphological methods in image segmentation to accurately extract the features of crack defects, but this algorithm requires high image quality, can only identify a few types of defects, and cannot meet the requirements of real-time detection.

The threshold-based image segmentation algorithm is simple and computationally efficient. The edge detection-based image segmentation method is fast, and the edge is accurate. The region-based image segmentation algorithm overcomes the defect of small image space segmentation associated with other segmentation algorithms. However, all these three methods have the disadvantage of poor antinoise performance. As a result, an image segmentation approach based on the entire convolutional neural network structure was created in this study. For transfer learning based on the pre-training network, the full convolutional neural network can be employed, which reduces training time and improves

model robustness and parallelism. In this study, we constructed a model based on the SegNet deep convolutional neural network to realize automatic segmentation and labeling of sewer defect areas in images.

## 4. Image Segmentation Model Based on Deep Learning

The SegNet network is an open-source image segmentation project developed by the University of Cambridge team, which separates the areas of objects in an image, such as cars, roads, pedestrians, etc., onto the pixel level. SegNet networks circumvent the issues of computational convolution and pixel block storage. They are similar to traditional neural networks in that they are trained end-to-end to classify sets of pixels. The difference is that the SegNet network is made up of a coding layer and a decoding layer.

On the basis of semantic division of the full convolutional neural network, the network achieves end-to-end pixel-level image segmentation by constructing a symmetric encoder–decoder structure, in which the encoder uses pooling for downsampling and the decoder uses inverse convolution to gradually recover the detail and spatial dimension of the original image.

### 4.1. Encoders–Decoder Construction

The SegNet network is a full convolutional neural network. The main components of the central structure are a coding network, a decoding network, and a pixel-level classification layer that is connected after the decoding network. The structure of the coding network is identical to the topology of the 13 convolutional layers of VGG16 [22], resulting in low-resolution features. It retains feature maps with high resolution by rounding off the coded output position of the fully connected layer. For pixel-level classification, the decoding network simply substitutes the pooling layer in the coding network with the upsampling layer and maps the low-resolution features onto the whole input image-level resolution feature map. Figure 1 shows the SegNet network structure.

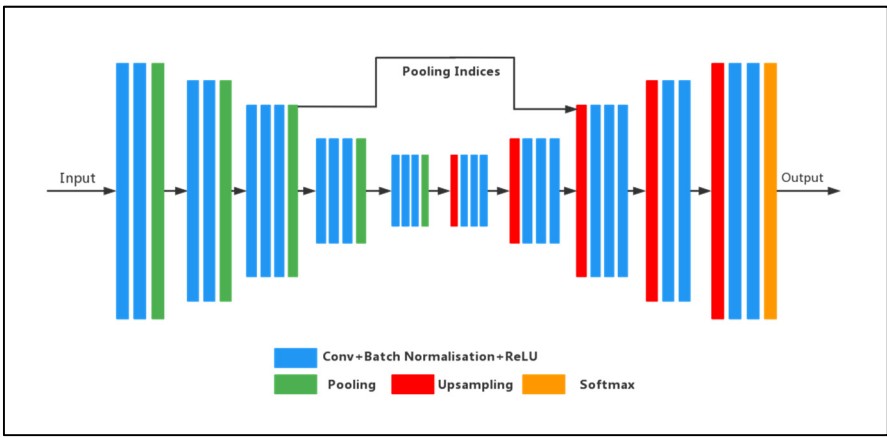

**Figure 1.** SegNet network structure.

In general, the network is divided into two parts. On the left is the encoding layer, in which the image is made smaller by pooling and the high-dimensional information and features in the graph are extracted. On the right is the decoding layer, which includes deconvolution and upsampling. Upsampling restores the original image size while deconvolution allows the image to be categorized and the features to be recreated. Finally, the softmax classifier is used to obtain the final segmentation map.

### 4.2. Pooling Indices

The aggregate of data for features at various locations in a particular region of an image is known as pooling. The SegNet network performs max pooling operations, which

means it is downsampled, and the corresponding index position is stored at the encoding time and upsampled at that position at the decoding time.

The three main functions of pooling are as follows: (1) The perceptual region is enlarged; one pixel can equate to an area in the original image; (2) due to translational invariance, pooling abstracts regional characteristics without regard for position; (3) reducing the difficulty and parameters of optimization.

Upsampling using a properly designed pooling index is utilized in the SegNet model's upsampling procedure, which lowers information loss caused by the pooling operation. In addition, the layer jumping connection is used to transfer low-layer features, increasing the amount of information included in the features. In conclusion, the model effectively improves the image segmentation rate.

### 4.3. Structural Analysis of the SegNet Network

The coding layer uses the VGG network as part of the coding. As can be seen from Figure 2, the subsample is divided into five parts, consisting of 13 layers [22]. The first and second parts are composed of two 3 × 3 convolution layers and a 2 × 2 pooling layer, and the third, fourth, and fifth parts are composed of three 3 × 3 convolution layers and a 2 × 2 pooling layer. Batch normalization is required for each result after each convolution operation and is triggered through activation functions. The pooling layer extracts the feature information in the image through the maximum pooling operation, saves its maximum index in the course of each pooling operation, such as the position information of the maximum feature value, and then releases its index during the upsampling process.

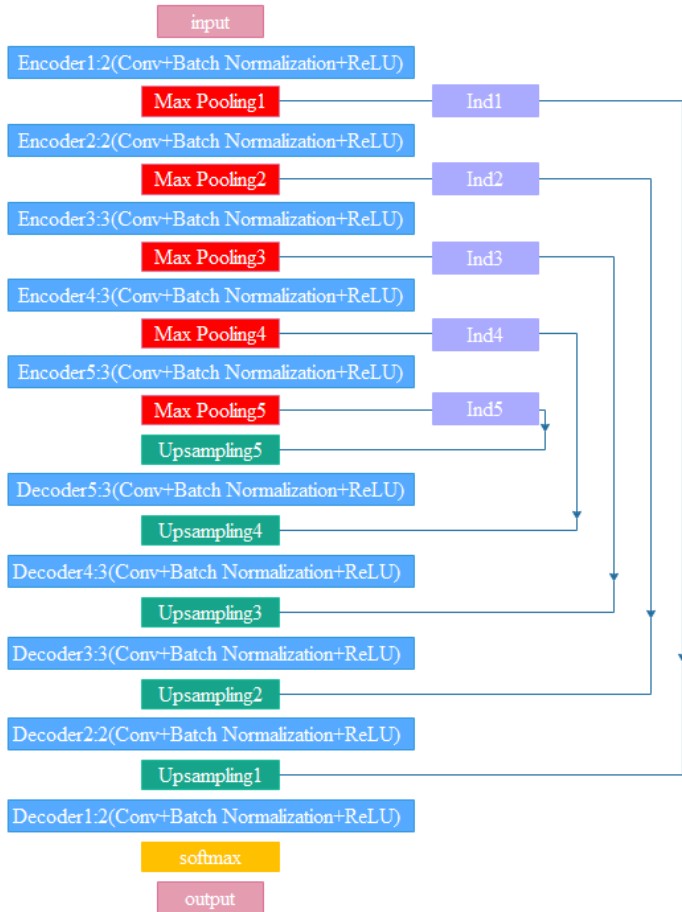

**Figure 2.** Detailed diagram of the SegNet network.

The decoding layer employs a mapping technique to convert a single high-order, low-resolution feature graph into a full-resolution feature map. The encoding layer records the largest index in the pooling operation through the encoding process. The decoding network uses a pooled index in the largest pooling layer to perform nonlinear upsampling on the image, as shown in Figure 3, which does not require learning [12]. As can be seen from Figure 2, the decoding process corresponds to the encoding process, and the decoding process is also divided into five parts. The first and second parts consist of a 2 × 2 deconvolution layer and two 3 × 3 convolutional layers. The third, fourth, and fifth parts are composed of a 2 × 2 deconvolution layer and three 3 × 3 convolutions, and finally each pixel is predicted in the obtained feature image by a softmax classifier. Not only can the encoder increase edge characterization, but it can also minimize the amount of network training parameters and speed up training.

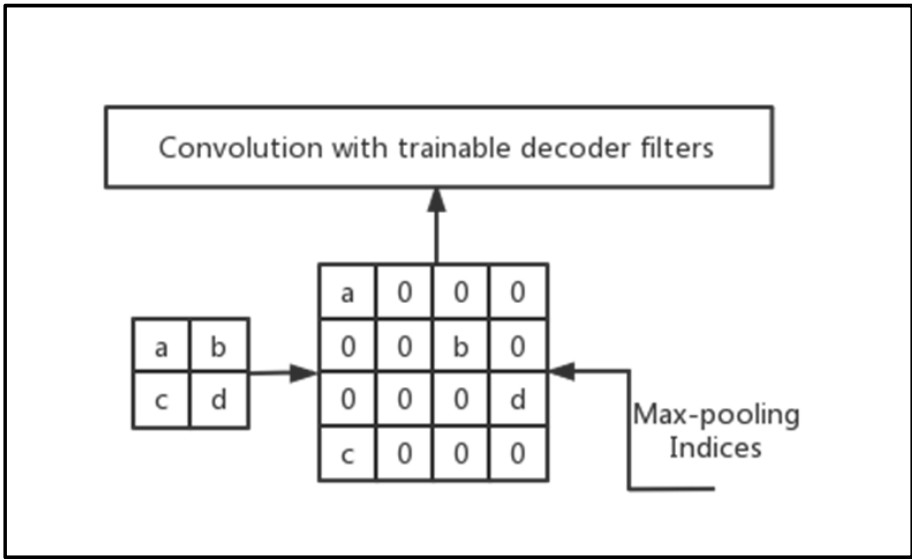

**Figure 3.** Upsampling schematic.

## 5. Experimental

### 5.1. Data Preparation and Training

Detection images of an underground pipe network in an old city in northwest China were collected, and 700 images taken by CCTV were selected as datasets. To simplify the analysis, only sediment, scum, corrosion, roots, mismatch, obstacles, and branch pipes were discriminated separately, as shown in Figure 4. These seven types of defects are obvious characteristics with a high probability of occurrence and a significant impact. The dataset image was processed in advance to improve training outcomes. A histogram equalization procedure was applied to the image, and the outcome is displayed in Figure 5. The primary purpose of histogram equalization was to average the histogram of the original image in order to increase the gray value range of the image, thereby enhancing the image contrast and optimizing the visual effect.

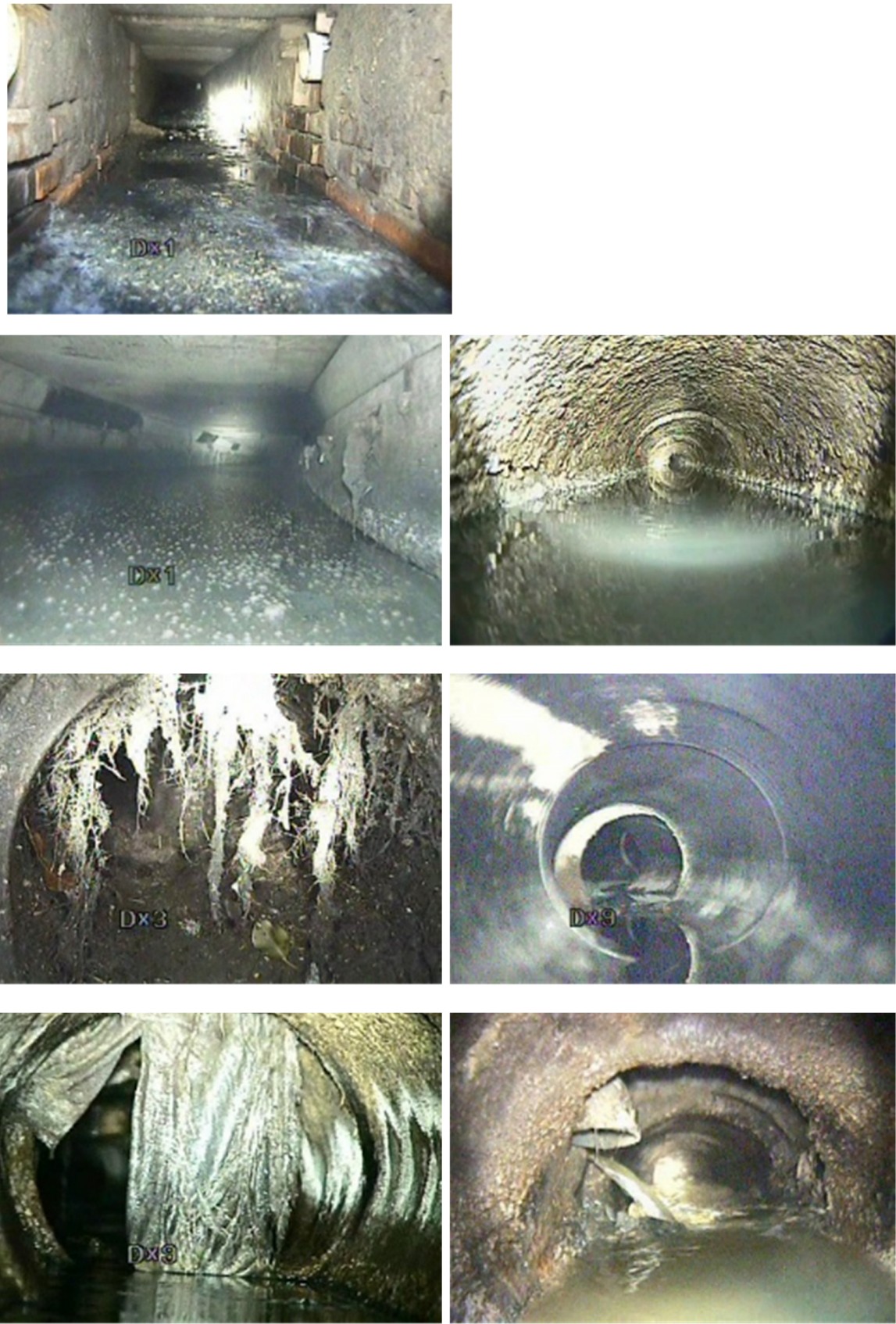

**Figure 4.** Classified defects in the experiments. From left to right and from top to bottom: sediment, scum, corrosion, roots, mismatch, obstacles, and branch pipes.

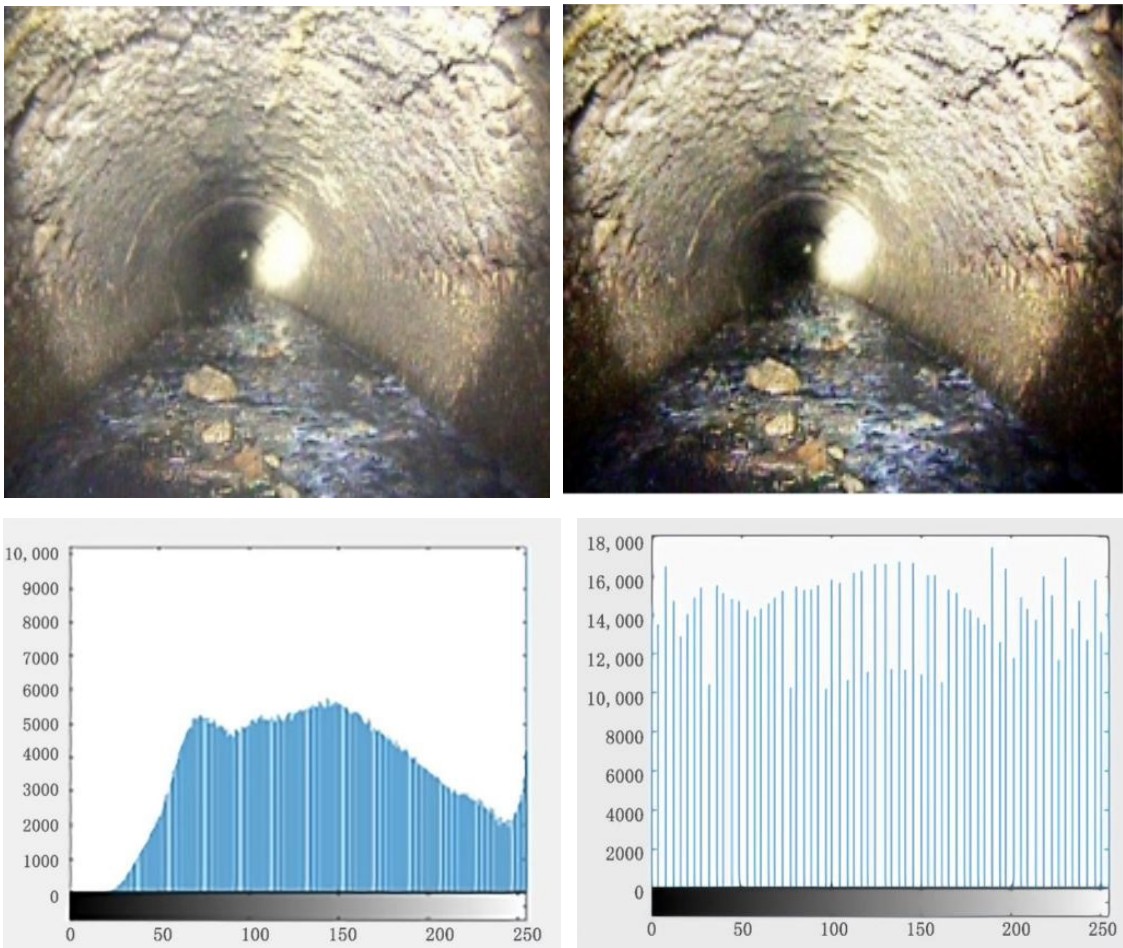

**Figure 5.** Histogram equalization of images. On the left is the original image and on the right is the balanced image.

After histogram equalization processing, a labeled image of each defect was created, the information of which is shown in Table 1. The above seven types of defects were marked by using Photoshop, and each picture was labeled with one major defect. To save training time and RAM, the original and labeled images were adjusted to a pixel size of 360 × 480. Figure 6 shows an example of the labeled images.

**Table 1.** Labeled image information table.

| Defect Type | Label Name | Label Pixel | Label Color |
|---|---|---|---|
| Corrosion | FS | 20 50 150 | 🟦 |
| Sediment | CD | 200 15 50 | 🟥 |
| Branch pipe | ZG | 100 50 200 | 🟪 |
| Scum | FZ | 160 250 160 | 🟩 |
| Roots | SG | 30 130 230 | 🟦 |
| Mismatch | CK | 240 140 240 | 🟪 |
| Obstacle | ZAW | 150 250 250 | 🟦 |

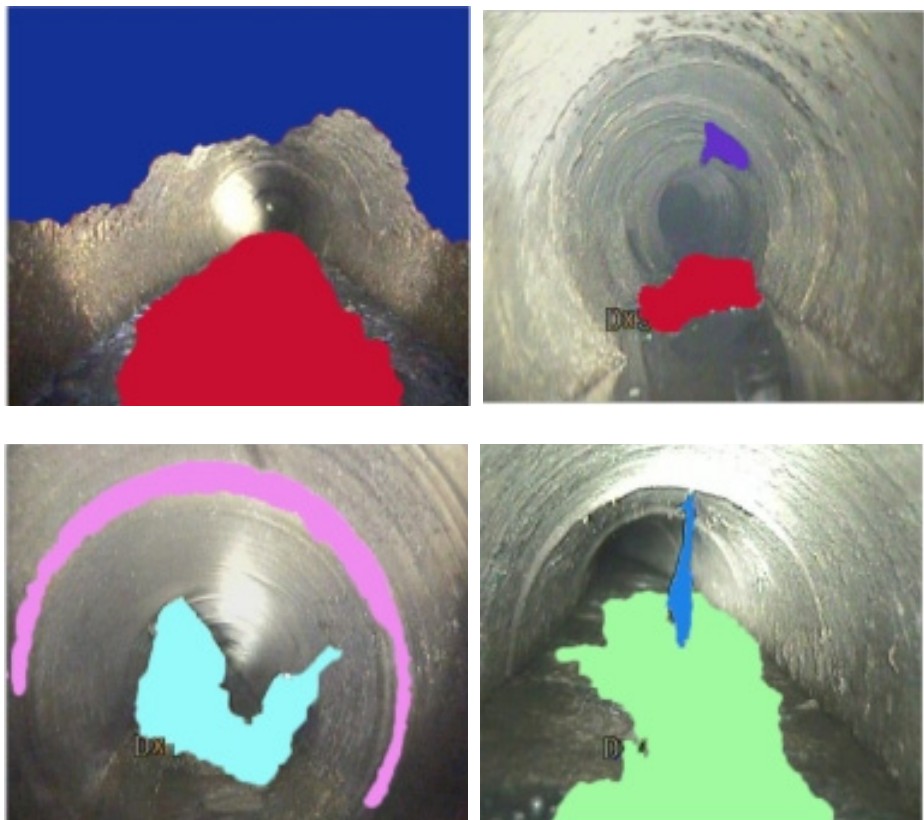

**Figure 6.** Examples of labeled images. The meanings of the labeling colors are shown in Table 1.

Pixel information from the labeled dataset is shown in Table 2 and Figure 7. It can be seen that the distribution of the various defect pixels is unbalanced. Because learning was oriented towards the dominant class, the imbalance may have had a negative impact on the learning process [23,24]. This problem was addressed by class weighting. The formula is shown in Equations (1) and (2), and the weights, after calculation, are shown in Table 3.

$$\text{imageFreq} = \frac{\text{PixelCount}}{\text{ImagePixelCount}} \tag{1}$$

$$\text{classWeights} = \frac{\text{median(imageFreq)}}{\text{imageFreq}} \tag{2}$$

**Table 2.** Statistical table of defect pixels.

| Name | Pixel Count | Image Pixel Count |
|------|------------|-------------------|
| FS | $2.99 \times 10^6$ | $1.08 \times 10^7$ |
| CD | $2.938 \times 10^6$ | $2.36 \times 10^7$ |
| ZG | $3.93 \times 10^4$ | $4.30 \times 10^6$ |
| FZ | $3.63 \times 10^6$ | $1.44 \times 10^7$ |
| SG | $2.53 \times 10^6$ | $1.69 \times 10^7$ |
| CK | $5.7633 \times 10^5$ | $1.46 \times 10^7$ |
| ZAW | $3.3152 \times 10^5$ | $4.32 \times 10^6$ |

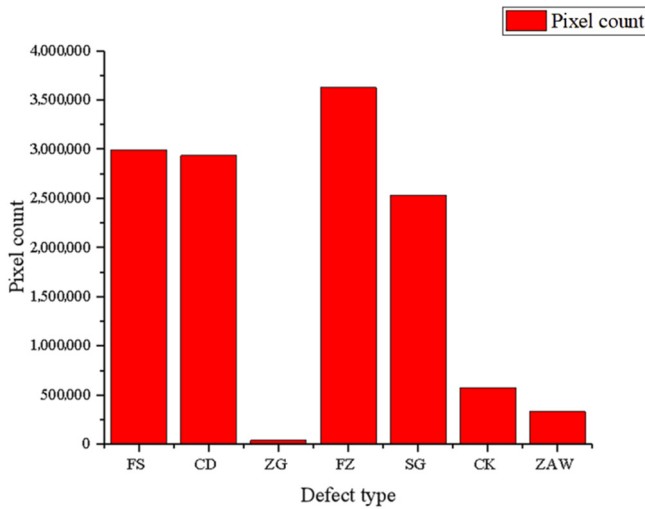

**Figure 7.** Statistical charts of defect pixels.

**Table 3.** Class weights of defect pixels.

| Name | Class Weights |
| --- | --- |
| FS | 0.45 |
| CD | 1.00 |
| ZG | 13.60 |
| FZ | 0.49 |
| SG | 0.83 |
| CK | 3.14 |
| ZAW | 1.62 |

After preprocessing was complete, the dataset was randomly divided into three subsets for training, verification, and testing. A total of 70% of the data was used for training, and the remaining 30% was divided into verification and test sets. To improve network accuracy and avoid overfitting, certain data augmentation methods, such as random left/right reflections and random left/right conversions, +/− 10 pixels, were used to provide more data. Six enhanced images were generated for each sample in the training set.

In the MATLAB platform, we used the vgg-16 network's initialization weights to create the SegNet network. The sewage dataset was trained using the SegNet network pre-trained with the CamVid dataset, with the stochastic gradient descent with momentum (SGDM) optimizer as the training technique. The specific parameter settings were as follows: momentum 0.9, initial learning rate 0.01, L2 regularization factor 0.0005, maximum number of epochs for training 90, and mini-batch size 4. As can be seen from Figure 8, as the number of iterations increases, the training loss tends to flatten out, and the model tends to converge.

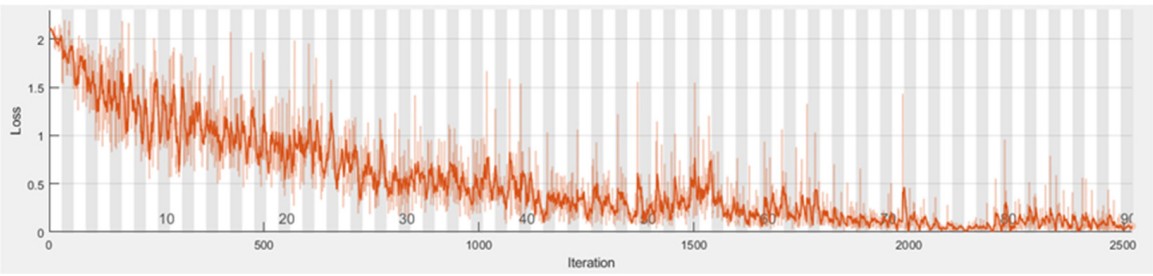

**Figure 8.** Statistical charts of defect pixels.

### 5.2. Results and Evaluation

After 90 iterations of training, test results were obtained, and a sample of these is shown in Figure 9. Comparing the test results with the labeled image, it is apparent that the network correctly identified the defect of scum. Only the main defect in the picture was marked in the label, but the defects not marked in the label were also identified by the network after training.

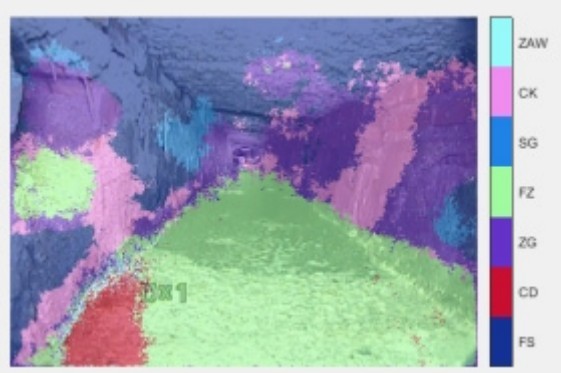

(**a**) Marking test charts.

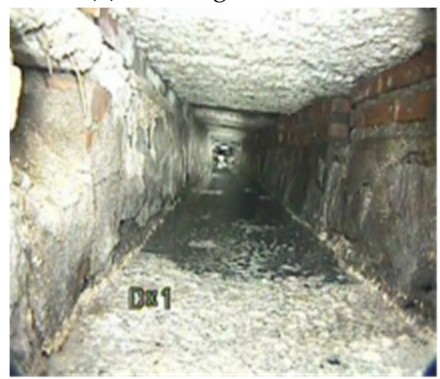

(**b**) Original image.

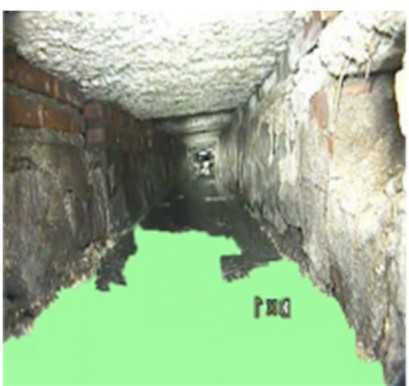

(**c**) Labeled image indicating the type of defect-FZ.

**Figure 9.** Comparison of single defect original image and segmented image. (**a**) represents the test chart of all marked colors, (**b**) an original map of the pipeline, and (**c**) the defect map of the segmentation mark, the defect is scum.

We further processed the image by processing the background to black, making it easier to identify the color of the segmented image in order to identify the type of defect in Figure 10. Figure 10 shows the original image and the segmented image, where the defects can be clearly seen.

For the segmented defect image, the accuracy, MIoU, and MeanBFscore were used as evaluation indicators.

Pixel accuracy (PA) is the ratio of the number of correctly segmented pixels to the total number of pixels, defined as in Equation (3). Mean pixel accuracy (MPA) is a simple improvement of PA. The proportion of correctly segmented pixels in each class is determined, and the average of all classes is calculated. The definition is shown in Equation (4):

$$PA = \frac{TP + TN}{TP + TN + FP + FN} \tag{3}$$

$$MPA = \frac{1}{k} \sum_{i=0}^{k} \frac{TP + TN}{TP + TN + FP + FN} \tag{4}$$

where TP, TN, FP, and FN represent the numbers of true positives, true negatives, false positives, and false negatives, respectively, and k indicates the number of categories.

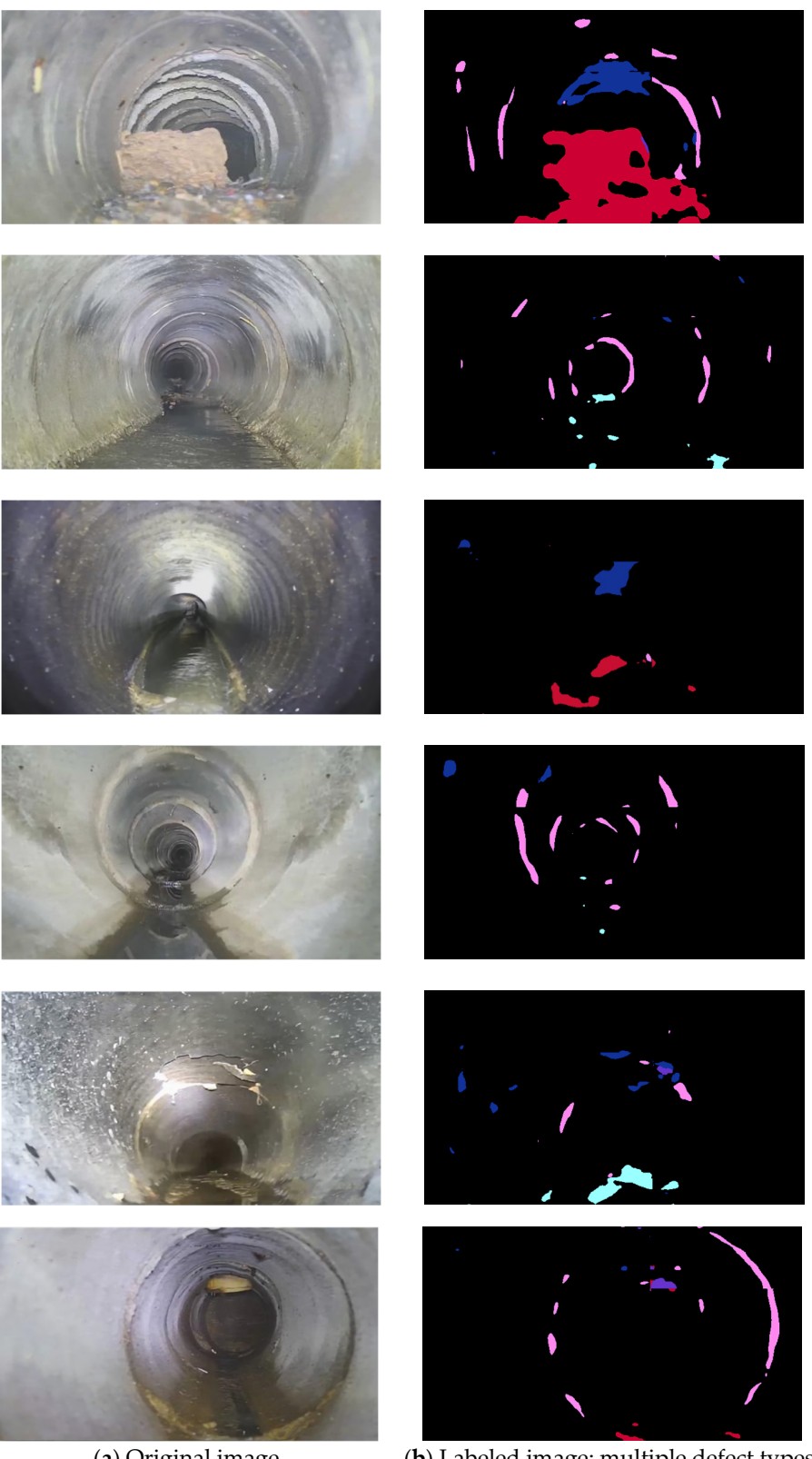

(**a**) Original image.　　　　　　　　(**b**) Labeled image: multiple defect types.

**Figure 10.** Comparison of the original image and the segmented image. The right segmentation image can accurately segment multiple defects in the pipe.

Table 4 presents the PAs and MPAs from the experiments performed in this study. The PAs were 82.77% and 79.59% on the validation and test sets, respectively, and the MPAs were 74.09% and 69.74% on the validation and test sets, respectively. Table 5 shows the PAs of each class in the test set. Among them, five types of defect segmentation (FS, CD, ZG, FZ, and SG) achieved 80% or more precision, but the effectiveness of CK and ZAW segmentation needs to be improved.

**Table 4.** Statistical Table 1 of recognition results.

| Indicators | PA (%) | MPA (%) | MIoU | BFScore |
|---|---|---|---|---|
| Valid | 82.77 | 74.09 | 0.61 | 0.75 |
| Test | 79.59 | 69.74 | 0.55 | 0.72 |

**Table 5.** Statistical Table 2 of recognition results.

| Name | PA (%) | IoU | BFScore |
|---|---|---|---|
| FS | 86.36 | 0.78 | 0.73 |
| CD | 86.27 | 0.68 | 0.69 |
| ZG | 79.46 | 0.64 | 0.55 |
| FZ | 81.64 | 0.72 | 0.62 |
| SG | 80.97 | 0.62 | 0.71 |
| CK | 47.09% | 0.29 | 0.81 |
| ZAW | 26.38% | 0.12 | 0.35 |

MIoU is a commonly used evaluation index for semantic segmentation. Its definition is given in Equation (5), where $k$ indicates the number of categories. It calculates the intersection and union ratio of two sets. In semantic segmentation, these two sets are the labeled image and the predicted image, and the MIoU reflects the degree of coincidence between the two sets. The closer the MIoU is to 1, the higher the coincidence degree and the higher the quality of semantic segmentation. MIoU is presented in Table 4, and the MIoU of each class in the test set is shown in Table 5. The MIoUs were 0.61 and 0.55, respectively, on the validation and test sets, and the MIoU of each class in the test set is presented in Table 5. The IoU of FS, CD, ZG, FZ, and SG was above 0.60, but the IoU of CK and ZAW was relatively low.

$$\text{MIoU} = \frac{1}{k} \sum_{i=0}^{k} \frac{\text{TP}}{\text{FP} + \text{TP} + \text{FN}} \tag{5}$$

Another widely used metric for semantic segmentation is the Boundary F1 (BF) score [25] (Equations (6)–(8)). The BF score is the contour matching score between the predicted segmentation and the true segmentation in the labeled set. The BFScores were 0.75 and 0.72, respectively, on the validation and test sets, and the BFScore of each class in the test set is presented in Table 5.

$$\text{Precision} = \frac{\text{TP}}{\text{TP} + \text{FP}} \tag{6}$$

$$\text{Recall} = \frac{\text{TP}}{\text{TP} + \text{FN}} \tag{7}$$

$$F_1\text{Score} = \frac{2 \times \text{Precision} \times \text{Recall}}{\text{Precision} + \text{Recall}} \tag{8}$$

The evaluation described above revealed that the FS, CD, ZG, FZ, and SG classes were easier to separate, but that it was difficult to separate CK and ZAW from the other classes. The reason for the poorer performance on ZAW was that there are many types of obstacles in the sewer and their division is usually based on object size, whereas the network used here is based on pixels.

## 6. Engineering Applications

Using a pipeline periscope and a CCTV inspection system, the sewers of 18 communities in a city in southern China were measured, focusing on drainage canals with a long construction time, a large diameter, and hidden dangers. A total of 700 images of pipes were taken. Figure 10 shows some of the pictures. The dataset photos were cropped to a uniform size of 360 × 480 pixels and subjected to a histogram equalization process.

The next stage was to determine whether the model is feasible in practice. The dataset was manually labeled before the trained network segmentation could be performed. Then, the model's automatic segmentation was compared with the manually labeled defects to evaluate network performance. The detailed information of the labels is shown in Table 1, and the labeling effect is shown in Figure 11.

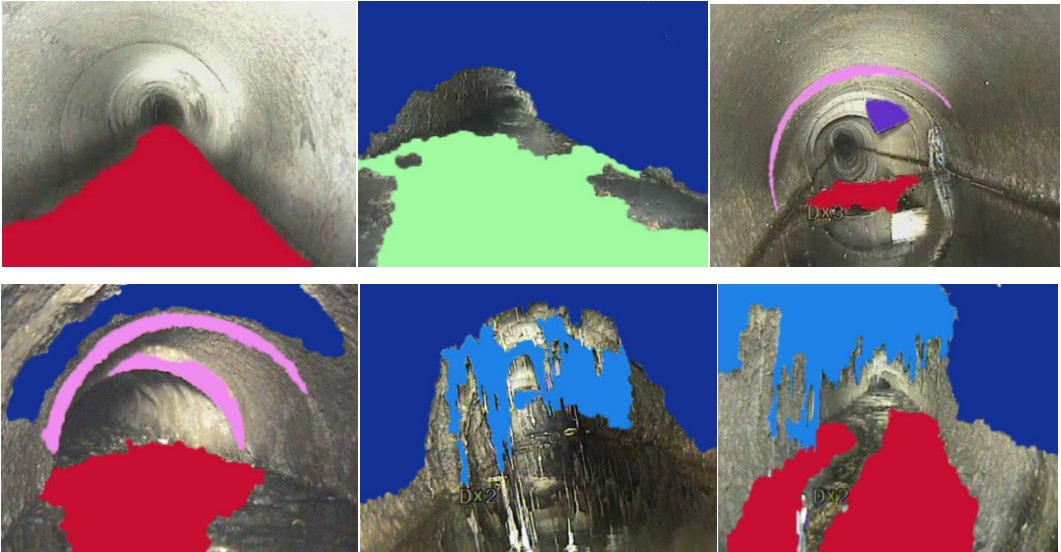

**Figure 11.** Statistical charts of defect pixels.

Finally, the original and processed images were fed into the trained deep learning network. Table 6 presents the evaluation results. The PA was 80.09%, the MIoU was 0.67, and the BFScore was 0.73. This shows that the network performed satisfactorily in an actual sewer inspection project.

**Table 6.** Evaluation results.

|  | PA (%) | MIoU | BFScore |
|---|---|---|---|
| Total | 80.09% | 0.61 | 0.73 |

## 7. Conclusions and Future Work

This paper proposes a method for automatically classifying and segmenting sewer defects based on deep convolutional neural networks. The SegNet network was used as the backbone network, and the CamVid dataset was pretrained and migrated to the collected pipeline defect image dataset.

To improve segmentation accuracy and reduce data imbalance, histogram equalization processing was first performed on the images in the dataset to enhance contrast. Then, a weighting method was used to balance defect pixels, and finally, an image data augmentation method was used to increase the number of training sets. Good performance was obtained on the verification and test sets, with PAs of 82.77% and 79.59% on the validation and test sets, respectively. When the trained network was applied to actual engineering project detection images, the PA was 80.09%, the MIoU was 0.67, the BFScore was 0.73, and the segmentation result was ideal, which demonstrates that the model is well suited for use in practical engineering.

A deep learning model was used for intelligent defect classification and region labeling in sewers, and good detection results were achieved, but the data volume was small, and the single machine operating speed was slow. Therefore, it will be necessary to further expand the sample base and improve segmentation effectiveness. In the present study, only seven defect types were classified and segmented. Future research may segment more kinds of defects. Comparative experimental analyses can be carried out in the future using a number of different networks. Moreover, FNC, DeepLab, and other networks can be used to classify and segment defects to obtain better performance.

**Author Contributions:** Conceptualization, M.H. and Q.Z. (Qin Zhao); methodology, H.G. and X.Z.; software, Q.Z. (Qinnan Zhao); validation, M.H., H.G. and Q.Z. (Qin Zhao); resources, M.H.; data curation, M.H. and X.Z.; writing—original draft preparation, Q.Z. (Qinnan Zhao); writing—review and editing, M.H. and H.G.; visualization, X.Z.; supervision, H.G.; project administration, M.H.; funding acquisition, Q.Z. (Qinnan Zhao). All authors have read and agreed to the published version of the manuscript.

**Funding:** This research was funded by National Natural Science Foundation of China, grant number 51878556 and Key Research and Development project of Shaanxi Province (No. 2022SF-197).

**Institutional Review Board Statement:** The study was conducted according to the guidelines of the Declaration of Helsinki, and approved by the Institutional Review Board.

**Informed Consent Statement:** Informed consent was obtained from all subjects involved in the study.

**Data Availability Statement:** The data used to support the findings of this study are available from the corresponding author upon request.

**Acknowledgments:** Thanks to all authors for their efforts in conducting this research.

**Conflicts of Interest:** The authors declare no conflict of interest.

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
