# Peer review of "Image Segmentation of a Sewer Based on Deep Learning"

_sustainability, doi:10.3390/su14116634_

Round 1

Reviewer 1 Report

To achieve pixel-level image segmentation of defect regions while clas
sifying pipe defects, this paper proposed a method based on deep convolutional neural networks. First, the image defect locations of seven typical defects were manually labeled to create the dataset. Then a model based on the SegNet network was used to label defect areas automatically in an image. The previously trained model from the CamVid dataset was applied to the pipeline image dataset. Finally, the model was applied to drainage pipe network images that were provided by periscope and closed-circuit television (CCTV) inspection cameras, and the pixel accuracy (PA) of image segmentation reached 80%. 
Comments are given as follows:
1. The topic should be modified to reflect the innovation of the algorithm.
2. Some grammatical errors appear in the abstract, the innovation cannot be detected by readers easily.
3. The title of Section 2 should preferably be changed to the related work.
4. The title of Section 2 should preferably be changed to image processing based methods, and title of Section 2.2 has the same problem.
5. The contents of Section 3 should not appeared in the paper because which is the basic image processing contents that can be found in related books.
6. Lack of comparative experiments in the experimental part so that advantage of the paper is difficult to prove.

Author Response

Thank you very much for your valuable comments, and my details of this revision are as follows:

  • I have made some changes to the subject matter.
  • I have made some modifications to the abstract to highlight the innovation of the algorithm in pipeline defects.
  • I revised the title of Section2 to related work.
  • I revised the title of Section2 to related work.
  • I removed the basic image processing content from Section 3.
  • In this paper, the comparison of image classification and image segmentation is added to verify the innovation, and the comparative trials in the experimental part are still under further study.Section 3.1 was added:Comparison of image classification and image segmentation.

Reviewer 2 Report

Dear Editor and Dear Authors,

The topic of this paper is very interesting and thank you for your trust to send me the manuscript to review it.

The manuscript presents a method based on deep convolutional neural networks to achieve pixel-level image segmentation of defect regions while classifying pipe defects. The authors stated that the pixel accuracy of image segmentation reached 80% and from the results it can be concluded that image segmentation and annotation technology based on deep learning is applicable to sewer defect detection.

The comments and suggestions to authors you can find below:

  1. The Abstract is satisfactory, however it is unusual to use abbreviations in it.
  2. The Introduction section is good, however I think it should contain the brief summary of the paper itself in the end of the section. This is an important point of view of all papers.
  3. The literature review contains several references from “arxiv.org”. It is unusual for high quality papers, since the articles from that website are not peer reviewed. These references can stay in the manuscript, however this section should be expanded with more valuable references from scientific literature, from high quality journals, since MDPI Sustainability is a high quality journal too with high impact factor.
  4. The expressions between lines 130-134 are written with Equation Editor? They are not looking good.
  5. The part with image segmentation should be expanded, since there new, very efficient algorithms that are not based on neural networks. The thresholding, edge detection and region-based segmentation are OK, but the new algorithms should be covered in brief too for the completeness od this subsection. Which literature was used for the image segmentation research? There are no references in this subsection?
  6. All the abbreviations in the paper should be explained.
  7. The Method part related to SegNet should be explained with more details.
  8. What software platform was used in experiments?
  9. Seven type of defects are mentioned. I think all these defects should be presented in separate images for the better clarification. It should be very clear what is the goal of the object detection. The image examples are not very clear due to the CCTV recording, and for readers is not obvious what defects are important for the algorithm.
  10. The Conclusion section is good.

The paper is very interesting and useful from the practical point of view. It has drawbacks that should be corrected and the paper should be revised.

Recommendation: Revision.

Author Response

Thank you very much for your valuable comments, and my details of this revision are as follows:

  • The abbreviation in the abstract has been deleted.
  • At the end of the introduction section, I added a brief summary of the paper itself.The proposed network is used to classify and segment various defects with image labels.The main goal is to meet the needs of large-scale sewage network survey work and realize intelligent defect detection.
  • The literature review contains some references from "arxiv.org" due to the incorrect format cited at the time that has been corrected to the correct reference format.
  • The expression between lines 130-134 is that the contents of Section3 have been deleted.
  • Section 3.1 was added some literature for image segmentation.
  • The abbreviations in this article are explained.The names of the label: FS, CD, ZG, FZ, SG, CK, and ZAW are custom names, as detailed in Table 1.
  • Interpretation of the SegNet related methods section is added in Section
  • In Section 5.1 illustrates the matlab software platform.
  • A variety of defective images were added in Section 5.2 to make the original and segmented image contrast more obvious.

Round 2

Reviewer 1 Report

The paper has been modified carefully according to the reviews' comments, I have no further question.